# Light Cross-Linkable Marine Collagen for Coaxial Printing of a 3D Model of Neuromuscular Junction Formation

**DOI:** 10.3390/biomedicines9010016

**Published:** 2020-12-26

**Authors:** Borja Sanz, Ane Albillos Sanchez, Bonnie Tangey, Kerry Gilmore, Zhilian Yue, Xiao Liu, Gordon Wallace

**Affiliations:** ARC Centre of Excellence for Electromaterials Science, Intelligent Polymer Research Institute, AIIM Facility, Innovation Campus, University of Wollongong, Squires Way, Wollongong, New South Wales 2500, Australia; bs540@uowmail.edu.au (B.S.); aas922@uowmail.edu.au (A.A.S.); btangey@uow.edu.au (B.T.); kerryg@uow.edu.au (K.G.); zyue@uow.edu.au (Z.Y.); xiaol@uow.edu.au (X.L.)

**Keywords:** 3D bioprinting, neural cell, skeletal muscle cell, neuromuscular junction

## Abstract

Collagen is a major component of the extracellular matrix (ECM) that modulates cell adhesion, growth, and migration, and has been utilised in tissue engineering applications. However, the common terrestrial sources of collagen carry the risk of zoonotic disease transmission and there are religious barriers to the use of bovine and porcine products in many cultures. Marine based collagens offer an attractive alternative and have so far been under-utilized for use as biomaterials for tissue engineering. Marine collagen can be extracted from fish waste products, therefore industry by-products offer an economical and environmentally sustainable source of collagen. In a handful of studies, marine collagen has successfully been methacrylated to form collagen methacrylate (ColMA). Our work included the extraction, characterization and methacrylation of Red Snapper collagen, optimisation of conditions for neural cell seeding and encapsulation using the unmodified collagen, thermally cross-linked, and the methacrylated collagen with UV-induced cross-linking. Finally, the 3D co-axial printing of neural and skeletal muscle cell cultures as a model for neuromuscular junction (NMJ) formation was investigated. Overall, the results of this study show great potential for a novel NMJ in vitro 3D bioprinted model that, with further development, could provide a low-cost, customizable, scalable and quick-to-print platform for drug screening and to study neuromuscular junction physiology and pathogenesis.

## 1. Introduction

Collagen is a major component of the extracellular matrix (ECM) and is the most abundant protein in the body [1,2,3]. In addition to providing structural support and tensile strength to tissues, collagen modulates cell adhesion, growth, and migration [3]. Due to its low immunogenicity and high biocompatibility, collagen has demonstrated biomedical applications in wound healing [4] as well as aiding periodontal [5] and bone and cartilage repair [6].

3D bioprinting is a fast growing technique that enables precise fabrication of complex constructs containing multiple cells and biomaterials [7,8]. Our group, for example, has developed 3D brain-like structures, consisting of discrete layers of primary neural cells [9] and fabricated implantable 3D pancreatic Islet-containing constructs [10], using bioprinting strategies. 3D printed constructs, such as a multi-material prosthetic hand [11] and dental implants [12], have wide clinical applications. Despite its natural biocompatibility, both unmodified collagen and the product of collagen denaturation, gelatin, form hydrogels with low tensile strength that are susceptible to rapid degradation [13,14] making them challenging biomaterials for use in 3D printing or implantation. To address this problem, chemical and physical methods of collagen and gelatin crosslinking have been developed, increasing the tunability of their mechanical properties. Enhancing crosslinking within the triple helical structure of collagen increases mechanical strength [15,16] and has been shown to decrease the rate of degradation [17,18]. Traditional chemical approaches however have been associated with cytotoxicity due to the persistence of crosslinking agents like glutaraldehyde within collagen [15]. On the other hand, physical methods such as thermal and UV crosslinking can result in collagen denaturation [19]. In addition, UV irradiation alone is ineffective as it does not result in a high degree of crosslink formation [19]. Photo-crosslinkable collagen or gelatin methacrylate are commonly used biomaterials due to their intrinsic biocompatibility and mechanically tunable characteristics [20]. Collagen methacrylate is formed by coupling of methacrylate groups to the amine-containing side chains of collagen. In the presence of light and a photo-initiator, a polymerization crosslinking reaction can then be triggered [21]. The desired degree of crosslinking can be controlled by the degree of methacrylation, the concentration of collagen, as well as the UV intensity and duration, and the concentration of the photoinitiator, making collagen methacryloyl a flexible biomaterial for tissue engineering and 3D bioprinting.

Currently, collagen is most commonly extracted from terrestrial animals such as cows, pigs, and poultry [22]. The use of these collagens as biomaterials in tissue engineering applications is limited due to risk of transmission of zoonotic diseases, such as foot and mouth disease, bovine spongiform encephalopathy, and avian influenza [23]. There are also religious barriers to the use of bovine and porcine products in many cultures [24]. Marine-based collagens have so far been under-utilized, however there has been increasing interest in them in recent years [1,25], particularly for use as biomaterials for tissue engineering [26]. Marine collagens offer an attractive alternative due to their low risk of pathogen transmission and lack of religious constraints. In addition, marine collagen can be extracted from fish waste products such as skin [27,28], bones [29,30], scales [31,32,33], and viscera [34,35]. Collagen has already been extracted and characterised for a broad range of marine species including jellyfish [36,37], cuttlefish [38], sea sponges [39,40], and numerous fishes [32,33,34,35,41,42,43,44,45,46]. In a handful of studies, marine collagen has also successfully been methacrylated to form collagen or gelatin methacrylate [20,26,47,48]. Given that an estimated 50% of commercial fish weight is currently discarded as non-consumable waste [49], fish industry by-products offer an economical and environmentally sustainable source of collagen.

Neural stem and progenitor cells have been shown to proliferate and differentiate in hydrogels containing porcine [50], bovine [51], and rat tail collagens [52,53]. To our knowledge however, investigations into the potential use of marine collagens as biomaterials for neural cell 3D tissue engineering has so far been limited to a study showing differentiation of induced pluripotent (iPS) cells to the specific lineage of dorsal cortical neurons in collagen extracted from the fish species Tilapia [54]. With respect to methacrylated marine gelatin or collagen, the only work that has been reported has been with fibroblast cells [20,26,47]. We aimed to assess the potential for methacrylated collagen from the common fish, Red Snapper, to be used as a biomaterial for 3D bioprinting of neural cells. For this study we used ReNcell VM and NSC-34 cell lines. ReNcell VM is a human neural progenitor cell line extracted from the ventral mesencephalon of a developing brain and immortalized via retroviral transduction using the retrovirus v-myc. NSC-34 is a murine motor neuron-like cell line formed from the fusion of motor neuron-enriched primary embryonic spinal cord cells and N18TG2 neuroblastoma cells that can be differentiated in vitro. NSC-34 cells were used for this project because these cells express many of the morphological and physiological characteristics of primary motor neurons, such as process extension, acetylcholine synthesis and storage, action potential support, neurofilament protein expression, contact establishment with cultured myotubes, and induction of myotube twitching [55,56]. Additionally, previous studies showed that NSC-34 cells were able to induce AChR clustering on myotubes when they were co-cultured [56]. Moreover, these cells, unlike other neuronal cell lines, can adhere specifically to the leucine-arginine-glutamine (LRE) motif of s-laminin, a neuromuscular synapse-specific basal lamina glycoprotein [57]. This feature provides evidence that these hybrid cells uniquely express motor neuron phenotypic characteristics. 

Our work included the extraction, characterization and methacrylation of Red Snapper collagen, optimisation of conditions for cell seeding and encapsulation using the unmodified collagen, thermally cross-linked, and the methacrylated collagen with UV-induced cross-linking. Finally, the 3D co-axial printing of neural and skeletal muscle cell cultures, as a model for neuromuscular junction formation, was investigated. 

## 2. Materials and Methods

### 2.1. Collagen Extraction and Purification

Collagen was extracted from the skin of Red Snapper, obtained from the Better Choice Fisheries, North Wollongong of NSW, Australia under sterile conditions. The skin was washed, cut into small pieces and suspended in 1 mL of 100 mM acetic acid (All reagents were from Sigma-Aldrich (Macquarie Park, NSW. Australia), if not differently specified) solution for every 60 mg of fish skin. The pH was adjusted to 2.5 by adding 1 M or 5 M HCl dropwise and the mixture reacted overnight with gentle stirring at 4 °C. The insoluble material was removed by centrifugation at 5500× *g* for 1 h. To separate the collagen from other proteins, 4.4 M of NaCl solution was added with intermittent gentle mixing, until the final concentration of NaCl reached 0.7 M. The precipitation was left to proceed overnight and the precipitate collected by centrifugation at 5000× *g* for 30 min. The collagen precipitate was solubilized in 500 mM acetic acid to stabilize. The product was dialyzed, firstly against 500 mM acetic acid at pH 2.5 overnight with gentle stirring. After filtration the sample was dialyzed against a 20 mM acetic acid solution for a further 24 h, replacing the solvent every 3–4 h, for a total of 3–4 times. Finally, the sample was dialyzed against cold deionized water for 1–2 h, before freeze drying and storing at −30 °C.

### 2.2. Methacrylation of Collagen

Collagen was dissolved in 20 mM acetic acid, to a final concentration of 2.5 mg/mL. The salt concentration was adjusted to 200 mM NaCl and the pH adjusted to 7.4 using 1 M NaOH at 0.023 × the volume of collagen. Methacrylic anhydride was added to target the residues of lysine and hydroxylysine with 52 µL of methacrylic anhydride added to every 100 mg of collagen. The reaction was carried out at 4 °C for 24 h and the product (ColMA) was dialyzed against 20 mM acetic acid for 48 h before freeze-drying. 

### 2.3. Collagen Hydrogel Formation

#### 2.3.1. Thermally Crosslinked Hydrogels

Collagen solutions were prepared at twice the final desired concentration by dissolving in 20 mM acetic acid. Final neutralized collagen solutions were prepared on ice and consisted of 500 μL collagen, 390 μL distilled water, 100 μL 10× PBS and 10 μL 1M NaOH per 1 mL of neutralized collagen. 200 μL of collagen was transferred per well of 48 well culture plates, and thermally cross-linked by incubation at 37 °C for 45 min.

#### 2.3.2. UV Cross-Linked Hydrogels

Methacrylated collagen hydrogels were prepared using the same protocol with the addition of lithium phenyl-2,4,6-trimethylbenzoylphosphinate (LAP) as photo-cross-linker to a final concentration of 0.1%. Methacrylated collagen was cross-linked by irradiating with an Omnicure 1000 UV lamp at 365 nm, at radiance of 12 mW/cm^2^ for 12, 25, or 50 s.

### 2.4. Fourier Transform Infra-Red (FTIR) Spectroscopy

A Shimadzu Fourier Transform Infrared Spectrophotometer IRPrestige-21 was used with a Miracle head attachment and IR Solution software. The freeze-dried foam version of collagen was used to create a homogeneous layer of material for analysis.

### 2.5. Circular Dichroism (CD) 

Samples were dissolved in 20 mM acetic acid to a concentration of 0.3 mg/mL. Samples were injected into a quartz cuvette with 1 mm path length in a Jasco J-810 spectropolarimeter. Ellipticity was recorded from 400 nm down to 190 nm in 1 nm intervals at constant temperatures (ranging from 5 °C up to 40 °C). Readings were taken in triplicate for each sample.

### 2.6. SDS-PAGE

Samples were dissolved in a solution of 20 mM sodium phosphate, containing 1.0% SDS and 3.5 M urea. NaOH was added dropwise to achieve a neutral pH. Samples were centrifuged for 5 min at 800× *g* to remove any undissolved particles. Samples (30 µL) were transferred to Eppendorf tubes and mixed with 5 µL of 6× sample buffer to give final concentrations of 62.5 mM Tris pH 6.8, 2% SDS, 10% glycerol and 0.01% bromophenol blue. Finally samples were heated at 100 °C for 5 min in a dry bath incubator and loaded into Bio-Rad Mini-PROTEAN TGX Stain-Free Gels (4–20%). 4 µL of Bio-Rad Precision Plus Protein Dual Colour Standards (Bio-Rad, Gladesville, NSW, Australia) and 10 µL of samples was loaded. An initial run at 60 V for 20 min was performed to stack all samples, then the gel was run at 120 V for 55 min. Gels were stained with 0.75 g/L (*w*/*v*) Coomassie blue R-250 (Bio-Rad, Gladesville, NSW, Australia) in 15% (*v*/*v*) methanol and 5% (*w*/*v*) acetic acid for 30 min and destained with 30% (*v*/*v*) methanol and 10% (*v*/*v*) acetic acid.

### 2.7. UV Rheology 

Rheological measurements were done with the TA Instruments AR-G2 Rheometer with a UV adapter, and an Omnicure series 1000 365 nm light source. Changes in both storage and loss modulus were determined, applying a 1% strain and frequency of 1 Hz with and without applied UV light. Light intensities were measured using a Silver Line UV Radiometer from CON-TROL-CURE. The geometry used was a 20 mm parallel plate with a 200 µm gap between the plate and the sample. 

### 2.8. 2D Surface Seeding of Cells 

NSC-34 cells (CELLutions, Toronto, ON, Canada) were cultured on both modified and unmodified fish collagen. Unmodified collagen coatings were prepared by coating wells of 48-well plates with 200 µL of 2.5% unmodified fish collagen. After allowing the collagen to thermally crosslink at 37 °C for 45 min cells were seeded to all gels at a density of 2 × 10^3^ cells/cm^2^. Methacrylated samples containing 2.5% collagen and 0.1% LAP were cross-linked by irradiating with a 365 nm UV lamp for 30 s at 12 mW/cm^2^ and seeded as above. Growth media consisted of a 50:50 mixture of high glucose Dulbecco’s Modified Eagle’s Medium (DMEM) and F12 Nutrient Mixture (F12) supplemented with 10% (*v*/*v*) fetal bovine serum (FBS, Bovogen, Interpath Services, Heidelberg West, VIC, Australia). Cells were maintained at 37 °C in a humidified 5% CO_2_ environment. After 24 h, wells under differentiation conditions were changed to media containing high glucose Dulbecco’s Modified Eagle’s Medium (DMEM), 2% horse serum and 100 U/mL penicillin/streptomycin. Media changes were performed every 2 days.

RenCell VM (Merck, Bayswater, Vic, Australia) neural progenitor cells were seeded to the gels at a density of 30 × 10^3^/cm^2^. After crosslinking, growth media consisting of (DMEM) with 2% B27 supplement, 10 ng/mL basic fibroblast growth factor (bFGF, Promega, Alexandria, Australia) and 20 ng/mL epidermal growth factor were added (EGF, Promega, Alexandria, Australia). Cells were maintained at 37 °C in a humidified 5% CO_2_ environment. After 24 h, wells under differentiation conditions were changed to the same media without growth factors. Media changes were performed every 2 days. 

### 2.9. Cell Encapsulation 

NSC-34 cells were encapsulated at 2 × 10^6^ cells/mL in unmodified or methacrylated fish collagen. An allotment of cell suspension, 200 μL, was deposited in each well of 48 well plates and cross-linked at UV irradiance of 12 mW/cm^2^ for 12, 25, or 50 s, or by thermal crosslinking at 37 °C for 45 min. Hydrogels were then soaked for 10 min in PBS to remove residual LAP, and then changed to either growth or differentiation culture media. Cultures were maintained at 37 °C in a humidified environment with 5% CO_2_. Media changes were performed every 2 days.

RenCell VM cells were encapsulated at 1 × 10^6^ cells/mL in unmodified fish collagen. Hydrogels were thermally cross-linked, soaked in PBS and transferred to culture media as described above. Media changes were performed every 2 days. Primary skeletal myoblasts isolated from 5 to 6 week old male mice and kindly donated by Anita Quigley (ACES, St. Vincent’s Hospital Melbourne) were encapsulated and differentiated in 5% methacrylated gelatin (GelMA, 82% degree of functionalization), provided by the Australian National Fabrication Facility (ANFF) Materials Node, University of Wollongong, Australia). The cells were cultured in HAM’s F10 media, supplemented with 20% FBS and 2.5 ng/mL bFGF and differentiated in DMEM supplemented with 1% horse serum.

### 2.10. 3D Bioprinting 

Coaxial printing utilised a customized printer (Figure 1) developed at the Translational Research Initiative for Cell Engineering and Printing (TRICEP, University of Wollongong, Australia). The dual syringes ink reservoirs have separate temperature control, allowing for independent temperature equilibration for each ink. The GelMA core was maintained at 16 °C while the methacrylated collagen shell was maintained at 18 °C, and printed at room temperature with the pressure around 140 to 200 kPa to ensure the formation of continuous lines [58]. 

We prepared the GelMA and neutralized fish ColMA solutions and suspended 10 × 10^6^/mL primary skeletal myoblasts or 2 × 10^6^/mL NSC-34 neurons in each solution, respectively. The suspensions were loaded into temperature-stabilized 3 mL syringes with diameters of 5 mm (Figure 1). The inner core was extruded at the centre of the print while the shell created a cylinder around it, producing a multi-material, concentric extrusion print. The extrusion was actuated independently by each module’s piston, creating a variable printing ratio. After optimization, we compared two printing ratios: 90:10 and 95:05 (core: shell). This allowed for good support provided by the GelMA core, whilst producing an intact layer of surrounding shell. The remaining printing parameters included x–y dimensions (10 × 10 mm), thickness of layers (0.6 mm), strand to strand distance of 1.5 mm, speed of print and flow (100 mm/min and 0.196 µL/mm, respectively). Scaffolds were irradiated for 12 s at 12 mW/cm^2^ (365 nm) and cultures maintained in myoblast differentiation media for 21 days, with media changes every second day.

### 2.11. Live-Dead Cell Viability Test

Staining of viable cells with 5 μM Calcein AM solution and dead cells with 1 μg/mL propidium iodide (PI) was used to assess the effect of encapsulation and UV treatment on cells. Viable cells take up Calcein and fluoresce bright green, whereas dead cells take up propidium iodide and fluoresce bright red. Cells were imaged using a Zeiss Axiovert 40 CFL inverted fluorescence and phase contrast microscope equipped with AxioVision software (SE64 Rel. 4.9.1, Zeiss Australia, North Ryde NSW, Australia).

### 2.12. Immunocytochemistry

For immunocytochemistry, 2D samples were fixed with 3.7% (*w*/*v*) paraformaldehyde (PFA) in phosphate-buffered saline (PBS) for 10 min at RT and subsequently blocked and permeabilized with 0.3% (*v*/*v*) Triton-X-100/10% (*v*/*v*) donkey serum (DS) in PBS for 1 h at RT. Alternatively, 2D samples were fixed and permeabilized with acetone/methanol (50:50) for 10 min on ice. In this case, blocking was achieved with 10% (*v*/*v*) DS/0.1% (*v*/*v*) Tween 20 in PBS for 1 h at RT. All samples were washed three times for 5 min in PBS. Samples were incubated overnight at 4 °C with primary antibodies diluted in 10% (*v*/*v*) DS in PBS. Primary antibodies were mouse (1:1000, Abcam, Melbourne, VIC, Australia) or chicken (1:1000, Merck-Millipore, Bayswater VIC, Australia), anti-β-III-tubulin, mouse anti-desmin (1:200, Novocastra- Leica biosystems, Mt Waverley, VIC, Australia), mouse anti-sarcomeric alpha actinin (1:200, Abcam, Melbourne, VIC, Australia). After incubation with the primary antibodies, samples were washed three times for 5 min with 0.1% (*v*/*v*) Triton-X-100 in PBS. AlexaFluor 594 conjugated donkey anti-mouse, AlexaFluor 488 conjugated goat anti-chicken and AlexaFluor 555 conjugated goat anti-mouse secondary antibodies (Thermofisher, North Ryde, NSW, Australia) were incubated for 1 h at 1:1000 dilution in 10% (*v*/*v*) DS in PBS at RT. AlexaFluor 555 conjugated bungarotoxin (1:500, Thermofisher, North Ryde, NSW, Australia) was also added to relevant samples at this step. 1 µg/mL of 4′,6-diamidino-2-phenylindole (DAPI, Thermofisher, North Ryde, NSW, Australia) in PBS was added for 5 min, followed by 3 × 5 min washes in PBS. Times for all steps were doubled for encapsulated cells in 3D samples, except for primary antibody incubations which were incubated overnight. A Zeiss Axiovert 40 CFL inverted fluorescence microscope equipped with AxioVision software was used for imaging. Immunostained bioprinted scaffolds were also imaged using a Leica TSC SP5 II confocal microscope equipped with LAS AF software (Leica Microsystems, Macquarie Park, NSW, Australia).

## 3. Results

Following the extraction, purification and freeze-drying procedures a translucent powder was obtained with a yield of 7% of the original dissolved and freeze-dried protein mixture. This purified product was subjected to the characterization studies described below.

### 3.1. FTIR Spectroscopy

FTIR spectra for bovine and marine collagen as well as denatured collagen (gelatin) were obtained (Figure 2). The absorbance at 1620–1670 cm^−1^ (amide I) associated with the stretching of the carbonyl groups (C=O bond), along the polypeptide backbone [59] and is a sensitive marker of the secondary structure of proteins [60]. The peaks observed at around 1700 cm^−1^ are normally attributed to collagen’s β-sheet structure, which is close to the theoretical point of random coil formation, usually assigned at around 1645 cm^−1^ [61,62]. Moreover, the amide III band at around 1238 cm^−1^ can also be seen, because of the NH bend, coupled with the CN stretch. Finally, the presence of both amide I and amide III bands indicates that the helical structure of collagen is intact. 

This was further assessed by comparison of the absorbance ratio between bands at 1235 and 1450 cm^−1^. It has been previously reported that the absorbance ratio I_1235_/I_1450_ of pure collagen is around 1 [63]. In the case of the bovine collagen used here the ratio was 1.04, while for marine-derived collagen the ratio was 1.02. Methacrylation of the marine collagen did not affect the FTIR spectra (Appendix A) and the ratio of absorbance bands at 1235 and 1450 cm^−1^ was maintained at 1.05.

When collagen is denatured it loses the triple helical structure, converting to gelatin. Gelatin has a I_1235/_I_1450_ cm^−1^ ratio of ≈0.69 [63], a I_1235/_I_1450_ cm^−1^ ratio of 0.67, in comparison with that of the extracted marine sample (1.02). This further supports the integrity of the collagen type I obtained from the extraction and purification protocol for the fish skin.

### 3.2. Circular Dichroism 

The triple helical structure of collagen results in a CD spectrum with a rotatory maximum at 221 nm, a minimum close to 205 nm, and a consistent crossover point at around 212 nm [64,65]. The CD spectra for the marine collagen, the methacrylated marine collagen compared to commercial bovine collagen and gelatin, are presented in Figure 3. The rotatory maximum and minimum point results are displayed in Table 1. The experimental results for both commercially available bovine collagen and the extracted marine collagen were close to the theoretical values and were unaffected by methacrylation of the marine collagen. Gelatin due to the loss of the triple helix lacks the rotatory profile.

### 3.3. SDS-PAGE Electrophoresis

SDS-PAGE electrophoresis was used to assess the purity of the extracted collagen. Collagen type I has a unique set of chains comprising its helical structure, each with its own specific molecular weight. The α-region comprising the two α-1 chains of the bovine collagen and the two batches of our extracted fish collagen is around 140 kDa (Figure 4). Moreover, the β-region shows the heavier α2 chain at around 250 kDa. Finally a γ-region is observed above the 250 kDa mark in both cases. This corresponds to the reported patterns for collagen type I found in the literature [66], where the α-region is localized between 100 and 150 kDa, while the β-region is close to 250 kDa and the γ-region at around 300 kDa. The denaturation of collagen into gelatin results in a blurred band, in which no distinctive regions are observed. The lack of any major blurring and the clarity of the component bands of our extracted fish collagen suggested that the purity of the extracted material was comparable to the commercially available bovine collagen. The clarity of bands in the methacrylated collagen sample confirms that no denaturation of the marine collagen occurred during the process of methacrylation.

### 3.4. Rheology

The storage modulus of methacrylated marine collagen at 2.5 and 1 mg/mL was tested during exposure to 12 mW/cm^2^ UV intensity for 12 s (Figure 5). Upon exposure to UV light the collagen gelled and an increase in storage modulus to 250 Pa (2.5 mg/mL) and 20 Pa (1 mg/mL) in less than 12 s was observed. Rapid crosslinking and gelation is critical for successful printing of this material.

### 3.5. Compatibility of Neural Cells with Marine Collagen

Collagen solutions were prepared and were either thermally or UV-crosslinked, as described in the Materials and Methods section. NSC-34 cells were seeded onto the collagen gels and cultured in growth media for 7 days, resulting in an increase in cell density. Those cultured in differentiation media showed morphological signs of differentiation by day 7 (Figure 6), i.e., the cells were flattened and elongated with short processes. NSC-34 cells encapsulated in fish collagen (Figure 7) failed to develop this differentiated morphology over a 14 day time period, but continued to proliferate. 

In both surface seeded and encapsulated cell experiments, ReNcell VM neural progenitor cells proliferated and induced contraction of the collagen gels over 7 days (60% contraction). This effect increased with cell density, as evidenced by the increased contraction with time as cells increased in number, as well as the contraction that was observed at early time points when cells were initially seeded at higher densities (e.g., cells encapsulated at 10 × 10^6^ cells/mL contracted the gels overnight, while at lower densities (2 × 10^6^ cells/mL–5 × 10^6^ cells/mL) contraction of the gels to one third of their original size occurred over a 7 day period. Encapsulation of cells at 1 × 10^6^ cells/mL allowed for long term culture (up to 21 days) without significant contraction of the gels. Cells cultured in differentiation media (both surface seeded and encapsulated) formed extensive branching neurite networks by 7 days (Figure 6 and Figure 8), indicative of differentiation. 

### 3.6. Live and Dead Staining 

When cells were surface seeded on marine collagen, live and dead staining at 1 and 7 days showed good cell survival for both NSC-34 and ReNcell VM cells in both growth and differentiation media with very few dead cells (Figure 6). The increase in cell number by 7 days in proliferation media was evident, as was the contraction of collagen gels with increasing densities of ReNcell VM cells. ReNcell VM cells in differentiation media formed interconnected neural rosettes by 7 days in 2D culture (Figure 6E,G).

Encapsulated cells in 3D culture, showed good cell survival for both NSC-34 and ReNcell VM cell types by 1 day and 7 days (Figure 7 and Figure 8) in both growth and differentiation media, however by 14 days, cell survival declined significantly. 

### 3.7. Compatibility of NSC-34 with UV-Cross-Linked Methacrylated Fish Collagen

The ability to produce UV-crosslinkable fish collagen paves the way for printing collagen based complex constructs. The NSC-34 cell line was selected to evaluate the cytocompatibility of the UV crosslinking process. The optimal UV-crosslinking condition was found to be 12 s at 12 mW/cm^2^. Longer cross-linking times, or higher irradiances in extrusion printed scaffolds was found to decrease cell viability by 24 h. NSC-34 cells were seeded to the surface of, and encapsulated in, UV-cross-linked methacrylated fish collagen gels and cultured for either 7 days (surface-seeded) or 21 days (encapsulated) in either growth media or differentiation media (Figure 9). Live and dead staining showed that cell viability was high and the cells continued to proliferate when surface seeded (by 7 days) or encapsulated (by 21 days).

### 3.8. 3D Coaxial Printing of Primary Myoblasts and NSC-34 Motor Neuron Cell Line

The neuromuscular junction was used as an example in this study to demonstrate the potential of building multiple cell type containing constructs, which is difficult to achieve via conventional 2D cell culture systems. Primary myoblasts and NSC-34 cells were successfully differentiated in 2D, both separately and in co-culture prior to setting up coaxial printing of skeletal myoblasts and neural cells. 

The optimum seeding density for 2D differentiation of NSC-34 cells was 2 × 10^3^ cells/cm^2^ whereas for skeletal myoblasts, which require a much higher 2D density for myoblast fusion, the optimum seeding density was 30 × 10^3^ cells/cm^2^. A range of differentiation media were tested against both cell types, with DMEM media supplemented with 1% horse serum and 1% P/S being chosen for co-cultures, based on the high degree of differentiation for both skeletal muscle and motor neuron cells in this media. For 2D co-culture, skeletal myoblasts were cultured at a density of 30 × 10^3^ cells/cm^2^ and differentiated for 3 days before seeding NSC-34 cells at a density of 2 × 10^3^ cells/cm^2^ and maintaining in the same media for 3–4 further days with media changes every 2 days.

Skeletal muscle cells and motor neuron cells were separately encapsulated in a range of marine collagen, and ColMA concentrations and in different GelMA formulations in terms of polymer concentration and degree of functionalization (DoF). The duration of cross-linking and UV irradiance were optimized in terms of resulting cell viability and 3D differentiation of the cells. The aim was to determine the conditions that were the most suitable to support motor neuron and muscle cell survival and differentiation separately, in 3D environments before proceeding to 3D bioprinting. Skeletal muscle differentiation in 3D was optimal at 10 × 10^6^ myoblasts/mL suspended in 5% (*w*/*v*) GelMA (82% DoF) and cross-linked with 0.1% LAP. Short cross-linking times, i.e., 12 or 25 s at an irradiance of 12 mW/cm^2^, resulted in high cell viabilities, as confirmed by live and dead staining at days 1, 7, and 14 after encapsulation and cross-linking. By 14 days in differentiation media, myofibers were spontaneously contracting in gels cross-linked for both 12 and 25 s, however, myotubes more homogeneously distributed in hydrogels crosslinked for 12 s. 

Achieving optimal conditions for differentiation of NSC-34 cells, both surface seeded and encapsulated in GelMA at a range of seeding densities, cross-linking conditions and degrees of functionalization was problematic in that cells continued to proliferate both in 2D and 3D. Differentiation was achieved on the surface of both methacrylated, and unmodified fish collagen using a range of cross-linking conditions. Live and dead staining performed at days 1 and 7 confirmed high cell viability β-III tubulin immunostainings were performed at day 7 to prove positivity for this neuronal marker. Due to the higher viability of encapsulated NSC-34 cells in methacrylated gels that were cross-linked for shorter periods of time, cross-linking of 2.5 mg/mL collagen gels for 12 s at 12 mWcm^2^ with cells at 2 × 10^6^/m, were utilised for subsequent 3D co-axial bioprinting.

Prior to bioprinting different printing parameters including pressure and speed of print, flow rate, temperature of the gels and receiving plates were tested to find optimum conditions to print 10 × 10 mm scaffolds of 3 layers in height. The duration of cross-linking at irradiance of 12 mW/cm^2^ needed to retain the integrity of scaffolds and retain cell viability at days 0, 7, and 14 after printing was determined to be 12 s. Immunostaining of skeletal muscle cells in gels cross-linked for 12 s after 14 days in culture using anti-α-sarcomeric actinin, confirmed that myoblasts had fused, forming multinucleated myofibers.

Once the optimized parameters for printing of both cell types had been achieved, the separate GelMA-laden and collagen-laden syringes were transferred to the nozzle (Figure 1) and the temperature of each module was adjusted for each bioink and allowed to stabilize for 15 min. The core module temperature was set to 16 °C to allow GelMA solution transition to a gel state and the shell module temperature was set to 18 °C. Single layer scaffolds were extruded at a flow rate of 0.196 μL/mm and a speed of 100 mm/min. Dimensions of the printed scaffolds were 10 mm × 10 mm × 0.6 mm (1.5 mm strand-to-strand distance). The GelMA core contained a high density (10 × 10^6^ cells/mL) of skeletal muscle cells, whereas the outer shell contained the NSC-34 motor neuron cell line at 2 × 10^6^ cells/mL in 2.5 mg/mL methacrylated fish collagen at 2 different ratios 95 (core):5 (shell) and 90:10. Both gels were cross-linked at the same time using UV light for 12 s at 12 mW/cm^2^. The scaffolds were incubated in skeletal muscle differentiation media which had been previously shown to support the highest percentages of differentiated myofibers and motor neuron-like cells. 

Within the coaxially-printed scaffolds the viability of both cell types was high for up to 21 days and both cell types were observed to differentiate, as evidenced by live and dead cell staining (Figure 10). Skeletal muscle cells were differentiating by 7 days in 3D culture and had formed thick, contracting myofibers that aligned with the direction of printing of the scaffolds by 14 days in culture. Live and dead imaging suggested the differentiation of both cells types in the scaffolds due to differences in morphology of both cell types. Neurite outgrowth and branching was observed, indicating neural differentiation in the 3D coaxial co-culture. Immunocytochemistry of the scaffolds after 21 days in culture (Figure 11), followed by confocal imaging confirmed the location of motor neurons in the shell component and myoblasts in the core component of the bioprinted structures as well as the differentiation of each cell type. Skeletal muscle cells were immunostained with the skeletal muscle protein α-sarcomeric actinin, while differentiated NSC-34 motor neuron cells were immunostained for β-III tubulin and neuro-muscular junctions were visualized using bungarotoxin staining. Myoblast differentiation into aligned myotubes was confirmed by α-sarcomeric actinin positivity (Figure 11B) and anti-β-III-tubulin showed that motor neuron-like cells were clustered and were in a different plane to the differentiated myotubes (Figure 11C). The presence of nodular α-bungarotoxin staining along myofibers in close proximity to motor neuron-like cells suggested that neuro-muscular junctions had formed by 21 days, however the presence of functioning neuromuscular junctions (NMJ) requires confirmation by electro-physiological studies.

## 4. Discussion 

There is an absence of a reliable in vitro NMJ model to study NMJ physiology and pathogenesis, as well as to provide a platform to study NMJ disease treatments and for drug screening. Bioprinting technology offers the opportunity to fabricate in vitro tissue models by precisely positioning biomaterials and cells to create relevant tissue models for drug screening and disease modelling [67]. This in vitro model has wide applications in clinical research such as in the development of personalised therapies for neurodegenerative diseases including amyotrophic lateral sclerosis (ALS) [68], investigations into the role of the synaptogenesis in neural degeneration [69], and as a drug testing platform. In this work, skeletal myoblasts and motor neuron cells were co-axially bioprinted to form a structured NMJ model in vitro. A bioink containing fish collagen with motor neuron cells was extruded as the shell and a bioink composed of myoblast-laden gelatin methacryloyl (GelMA) solution was extruded as the core component of bioprinted hydrogel strands. Considering that myoblasts and motor neurons are surrounded by different microenvironments in situ, this specific organization was used to support the adjustment of each bioink to each particular embedded cell type while combining them into a single organized structure. It was hypothesized that the motor neuron cells would extend neurites from the shell layer towards the periphery of the core layer, where skeletal myoblasts had been shown to localize [70], establishing motor neuron-skeletal muscle contacts and forming NMJs. This was confirmed by immunostaining and confocal microscopy of the printed construct, which also revealed the alignment of sarcomeric-actinin expressing myofibers, in agreement with the alignment and enhanced differentiation of skeletal muscle myofibers in 3D extrusion printed constructs that has been reported previously [71].

We have demonstrated the capacity of marine collagen extracted from Red Snapper fish skin to support neural cell survival and differentiation. While so far limited, some preliminary work has previously been done, showing improved neural cell culture conditions in collagen composites. Schuh, C.M., et al. for example, engineered neural tissue from collagen-fibrin composites which supported Schwann cells and promoted neurite outgrowth in vitro [72]. Further work needs to be done to investigate the use of marine collagen composites for 3D neural cell culture, opening the way to progress to the incorporation of marine collagen in more complex encapsulation systems. Marine-based collagens offer an attractive alternative to commonly used terrestrial animal based material, however, as a natural product from animals, batch-to-batch variation is unavoidable. Genetically engineered collagen may provide an alternative that overcomes this limitation [73]. Ideal biomaterials should mimic the properties of natural tissue. The ECM of the brain contains a high proportion of glycosaminoglycans and a smaller fraction of collagen and other fibrous proteins compared to other tissues [74]. Given this, collagen composites incorporating marine collagen with other materials, are likely to provide more suitable environments for neuronal modelling.

It is clear that modifications need to be made to improve the physical characteristics of fish collagen hydrogels in order to improve the printability of the scaffolds. We improved the mechanical properties of the fish collagen via methacrylation and UV cross-linking, allowing for 3D bioprinting of neural cell-laden gels alone, and as coaxial scaffolds mimicking physiological systems by bringing differentiating motor neuron-like cells into close proximity with differentiating skeletal myoblasts, contained within a different material (GelMA) that had been optimized to allow skeletal myofiber formation. The increased differentiation of motor neuron-like cells in proximity to skeletal myofibers in the coaxial scaffolds has parallels in developmental biology where the presence of skeletal myoblasts is known to enhance the development of neuro-muscular junctions [75,76].

In addition to its composition, the mechanical properties of a biomaterial influences cell proliferation, survival, migration and differentiation [77]. Biomaterials for neural cell culture should aim to mimic the stiffness of natural brain tissue, which is relatively low, ranging from approximately 100 to 1500 Pa. Our methacrylated fish collagen fell within this range, with a storage modulus of 250 Pa for the 2.5 mg/mL gel under our cross-linking conditions, and parallels the work of Iwashita et al. 2009, who produced gels derived from Tilapia fish collagen which they chemically cross-linked with 1-ethyl-3-(3-dimethylaminopropyl)-carbodiimide and *N*-hydroxysuccinimide in order to mimic brain tissue stiffness [54]. 

In this study the feasibility of co-axially bioprinting a NSC-34 motor neuron-laden collagen bioink and a skeletal myoblast-laden GelMA bioink to form a structured NMJ model in vitro was demonstrated. Co-axially bioprinting motor neuron-like cells and skeletal myoblasts in different bioinks allowed us to tune conditions for the differentiation of each cell type separately, while combining them into a structure that more closely captured the in vivo scenario in which the different cell types are encapsulated in different microenvironments. Overall, the results of this study show great potential for a novel NMJ in vitro 3D bioprinted model that, with further development, could provide a low-cost, customizable, scalable and quick-to-print platform for drug screening and to study neuromuscular junction physiology and pathogenesis. 

## Figures and Tables

**Figure 1 biomedicines-09-00016-f001:**
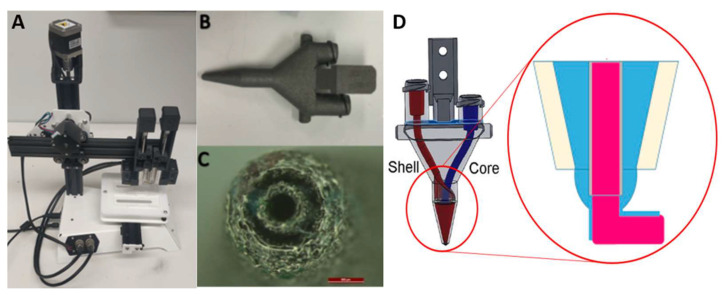
Custom made coaxial BioScribe 3D printer developed at TRICEP (**A**); 3D printer coaxial nozzle that allows for two syringes to be attached (**B**); Nozzle tip, comprised of an inner and outer orifice for the extrusion of the core and the shell components of the filaments, respectively. The scale bar represents 500 μm (**C**); Schematic drawing showing the channels inside the nozzle, being the core (red) used to extrude the supporting GelMA, while the shell (blue) is a softer coating of NSC-34 cells encapsulated in ColMA (**D**).

**Figure 2 biomedicines-09-00016-f002:**
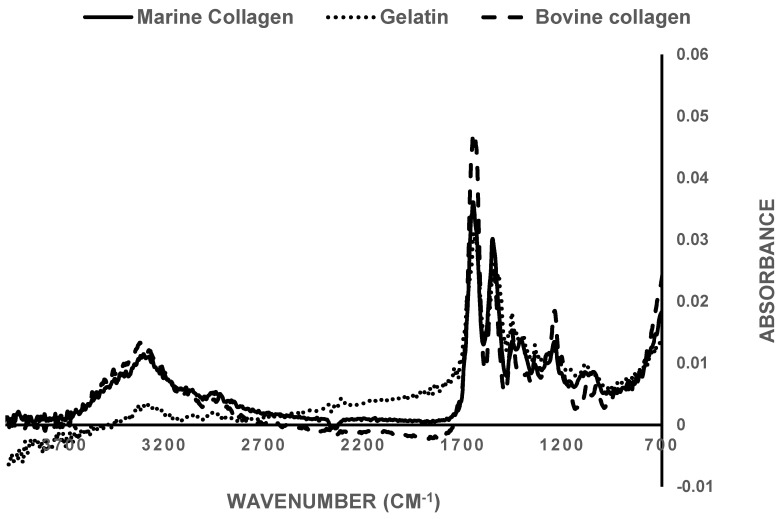
Comparison of FTIR spectra of extracted marine-derived collagen (solid line), commercially available bovine collagen (dashed line) and commercial gelatin (dotted line). Samples were tested as freeze-dried foams at RT.

**Figure 3 biomedicines-09-00016-f003:**
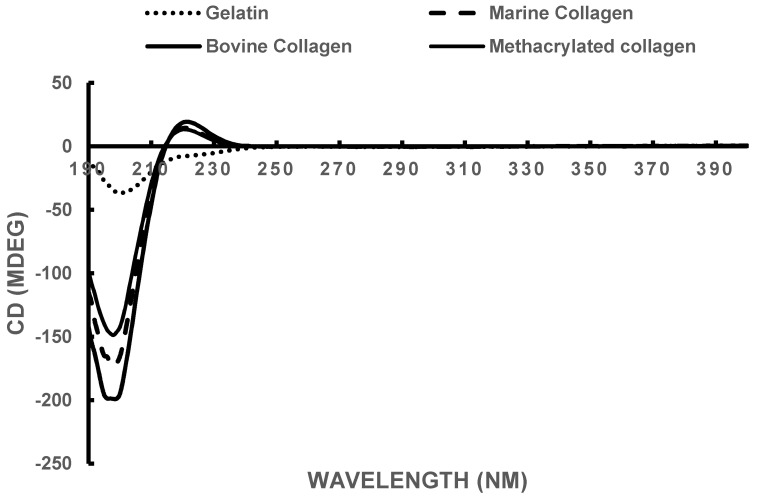
Circular dichroism spectra, taken at 5 °C. All samples were dissolved in 20 mM acetic acid and diluted to a final concentration of 0.3 mg/mL.

**Figure 4 biomedicines-09-00016-f004:**
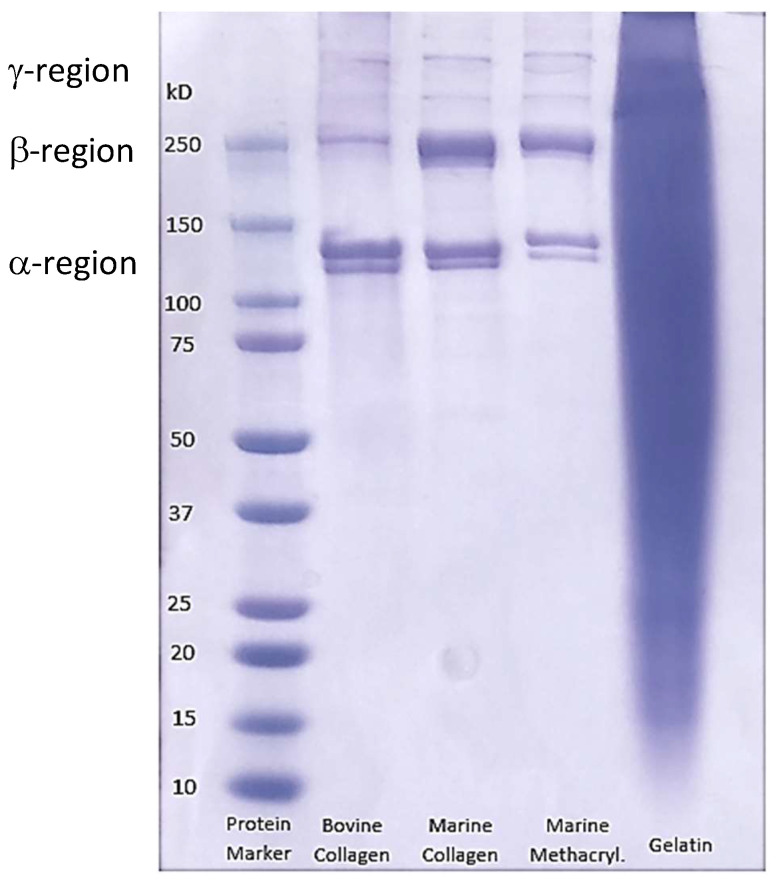
SDS-PAGE of commercially available bovine collagen and gelatin, compared with our extracted marine collagen, before and after methacyrylation.

**Figure 5 biomedicines-09-00016-f005:**
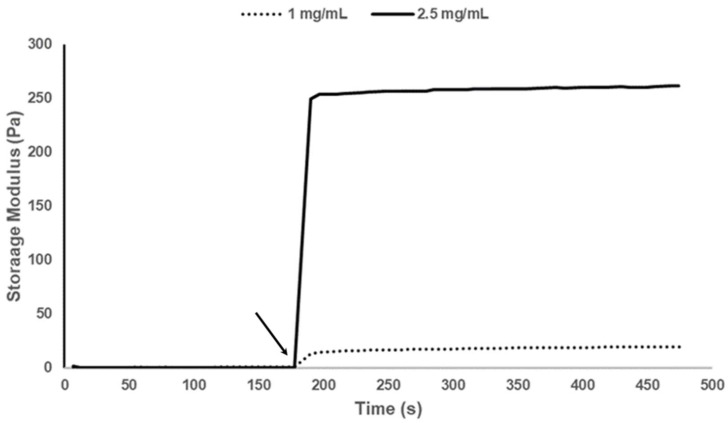
Storage moduli of both 2.5 mg/mL and 1 mg/mL methacrylated collagen in the presence of 0.1% LAP cross-linker. The arrow indicates the point at which the UV light was switched on. The samples were irradiated at 12 mW/cm^2^ for 12 s.

**Figure 6 biomedicines-09-00016-f006:**
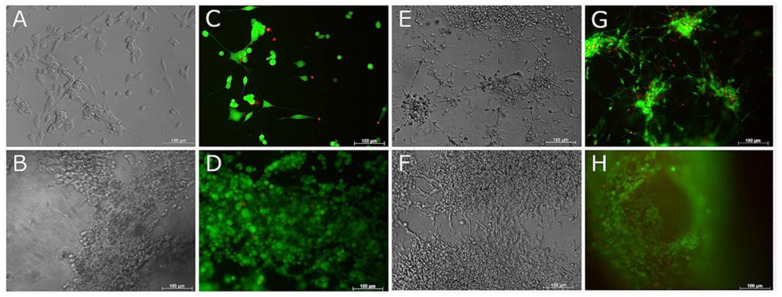
Live and Dead and phase contrast images of NSC-34 and RenCell VM neural progenitor cells surface seeded to 2.5/mL fish collagen. Phase contrast (**A**,**B**,**E**,**F**) and live and dead (**C**,**D**,**G**,**H**) images taken of NSC-34 (**A**–**D**) and RenCell VM (**E**–**H**) at day 7. Top and bottom rows correspond to cells cultured with differentiation media and growth media, respectively. Scale bars represent 100 µm.

**Figure 7 biomedicines-09-00016-f007:**
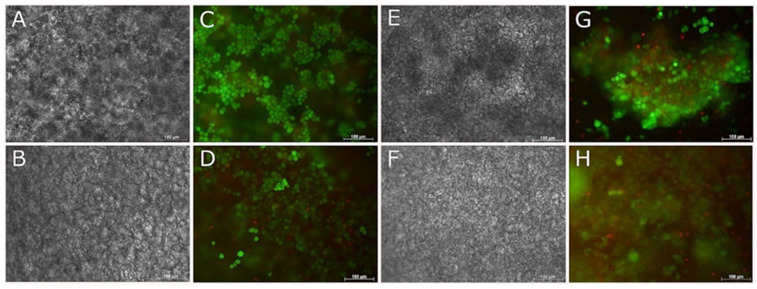
Live and Dead and phase contrast images of NSC-34 cells encapsulated in 2.5/mL fish collagen. Phase contrast (**A**,**B**,**E**,**F**) and live and dead (**C**,**D**,**G**,**H**) images taken of NSC-34 at 7 (**A**–**D**) and 14 (**E**–**H**) days. Top and bottom rows correspond to cells cultured with differentiation media and growth media, respectively. Scale bars represent 100 µm.

**Figure 8 biomedicines-09-00016-f008:**
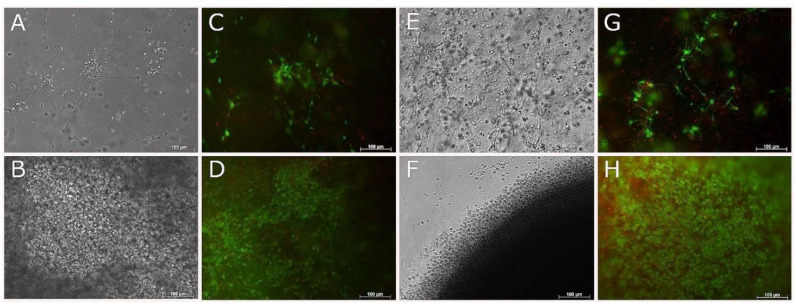
Live and Dead images of RenCell VM neural progenitor cells encapsulated in 2.5/mL fish collagen. Phase contrast (**A**,**B**,**E**,**F**) and live and dead (**C**,**D**,**G**,**H**) images taken of RenCell VM at 7 (**A**–**D**) and 14 (**E**–**H**) days. Top and bottom rows correspond to cells cultured with differentiation media and growth media, respectively. Scale bars represent 100 µm.

**Figure 9 biomedicines-09-00016-f009:**
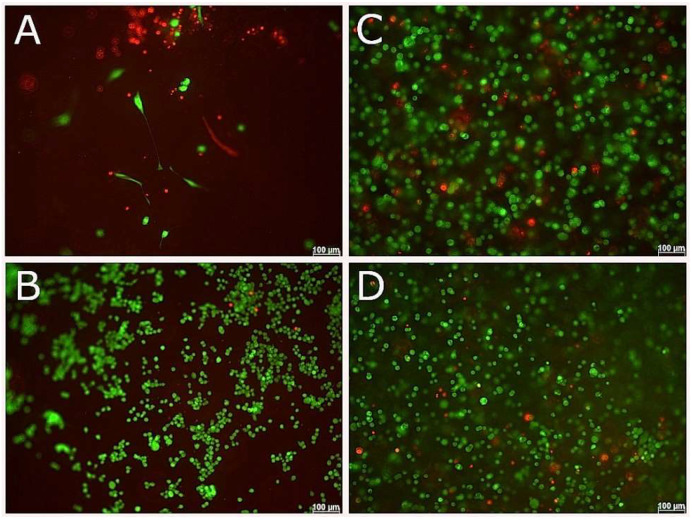
Live and Dead images of NSC-34 cells encapsulated in or surface seeded to 2.5/mL fish collagen UV crosslinked for 12 s at 12mW/cm^2^. Live and dead images taken of NSC-34 surface seeded to methacrylated fish collagen at day 7 (**A**,**B**) and encapsulated in methacrylated fish collagen at day 21 (**C**,**D**). Top and bottom rows correspond to cells cultured with differentiation media and growth media, respectively. Scale bars represent 100 µm.

**Figure 10 biomedicines-09-00016-f010:**
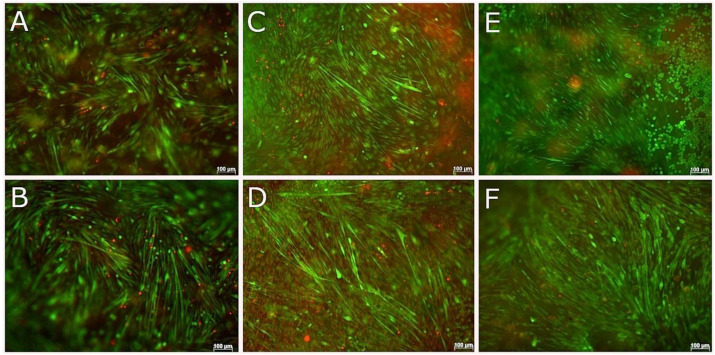
Live and Dead stained cells in coaxially printed scaffolds irradiated for 12 s at 12mW/cm^2^. NSC-34-laded collagen was extruded as the shell component and myoblast-laden GelMA was extruded as the core component of the bioprinted strands. Two ratios were used for extrusion, 95% core (5% GelMA)/5% shell (2.5 mg/mL methacrylated collagen) (**A**,**C**,**E**) and 90% core (5% GelMA)/10% shell (2.5 mg/mL methacrylated collagen) (**B**,**D**,**F**) at days 7 (**A**,**B**), 14 (**C**,**D**), and 21 (**E**,**F**). Scale bars represent 100 µm.

**Figure 11 biomedicines-09-00016-f011:**
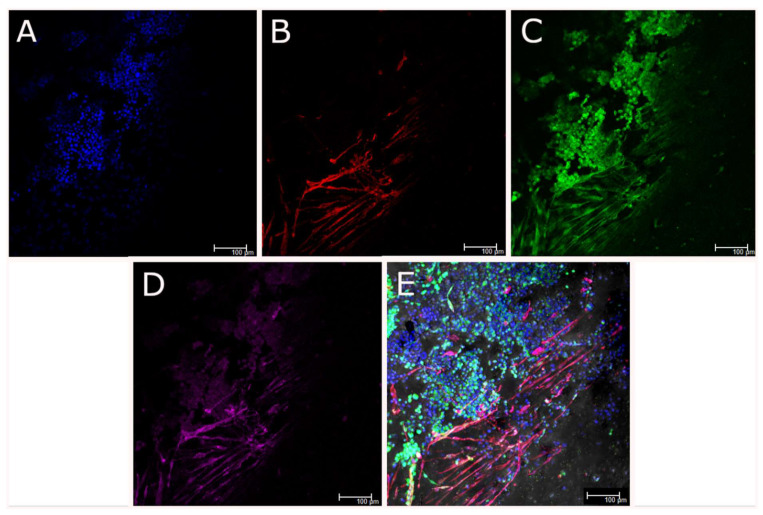
Coaxially-printed skeletal muscle and motor neuron cells, fixed and immunostained for nuclei (DAPI, blue, **A**), α-sarcomeric actinin (red, **B**), βIII tubulin (green, **C**) and neuromuscular junctions (α-bungarotoxin, magenta, **D**). Image **E** is an overlay of **A**,**B**,**C** and **D E**. Scale bars represent 100 µm.

**Table 1 biomedicines-09-00016-t001:** CD data for the extracted marine collagen, compared with commercial bovine collagen and gelatin. The theoretical values are also listed.

Sample	Rotatory Minimum (nm)	Rotatory Maximum (nm)	Crossover Point (nm)
Bovine Collagen	198	222	215
Marine Collagen	198	220	214
Methacrylated Marine Collagen	198	220	214
Gelatin	199	-	-
Theoretical Collagen [64]	205	221	212

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
