# Peer review of "Light Cross-Linkable Marine Collagen for Coaxial Printing of a 3D Model of Neuromuscular Junction Formation"

_biomedicines, 2020, doi:10.3390/biomedicines9010016_

Round 1

Reviewer 1 Report

the authors present an excellent study on the feasibilty and viability of 3-D bioprinting cell laden hydrogels fabricated in part with Red Snapper collagen. The authors have appropriately filled a knowledge gap via the use of the Red Snapper collagen as an alternative to collagens that may be out of the question for some cultures. The fact that the SDS Page results indicate no denaturation of marine collagen during methacrylation is a promising result. The introduction, materials/methods, and presentation of results is clear and professional.

I recommend acceptance after minor adjustments:

Content:

  • The authors have developed a very interesting customized coaxial 3-D bioprinter, however they did not discuss the pressures and types of force that the cells undergo during the extrusion process. Please include the print pressure used in the materials/methods section.

Formatting:

  • Sec 2.10 - The first paragraph appears to be centered rather than justified, please consider fixing.
  • Line 440 - "30x 10^3 cells/cm^2" - the spacing of the "x" should match the other times this is used ("30x10^3")
  • Line 472 - "...had fused ,forming multinucleated..." - the comma spacing should be fixed

Reviewer 2 Report

The manuscript topic is actual and the paper has merit. It could be attractive, adequate and interesting for the BIOMEDICINES journal Readers. However there are some concerns that authors should address in order to have a final more complete and attractive paper.

Biomaterials and Sea are currently developing a close relationship and new opportunities can be found its application in the clinical and experimental daily practice. It would be interesting if Authors may also underline the limitation of the value of the study, and the possible implication reflected on those materials

The authors application field is the neuromuscular one and it is pretty hard to realize all the features of this such anatomical district as the complexity of Physiology.

However, at this stage the paper seems to be high specific and directed to a short group of researchers such as not clinicians. Please emphasize possibile clinical role of the study results adding some references about next or translational application of the study, and its scientific rationale. Underline the educational message for the Readers

Some suggestions as References are for increasing the first part of the paper in order to better present the 3D printed technology. Introduction section is scientifically weak and should cover a more wide topic. 
Please increase the introduction section with some papers like the following. 

Barazanchi, A.; Li, K.C.; Al-Amleh, B.; Lyons, K.; Waddell, J.N. Mechanical Properties of Laser-Sintered 3D-Printed Cobalt Chromium and Soft-Milled Cobalt Chromium. Prosthesis 2020, 2, 313-320

Otte, A. 3D Computer-Aided Design Reconstructions and 3D Multi-Material Polymer Replica Printings of the First “Iron Hand” of Franconian Knight Gottfried (Götz) von Berlichingen (1480–1562): An Overview. Prosthesis 2020, 2, 304-312

Fiorillo, L.; Leanza, T. Worldwide 3D Printers against the New Coronavirus. Prosthesis 2020, 2, 87-90
